# Physiological and Genetic Basis of High-Altitude Indigenous Animals’ Adaptation to Hypoxic Environments

**DOI:** 10.3390/ani14203031

**Published:** 2024-10-19

**Authors:** Pengfei Zhao, Shaobin Li, Zhaohua He, Xiong Ma

**Affiliations:** 1Faculty of Chemistry and Life Sciences, Gansu Minzu Normal University, Hezuo 747000, China; zhaopf@st.gsau.edu.cn; 2Gansu Key Laboratory of Herbivorous Animal Biotechnology, Faculty of Animal Science and Technology, Gansu Agricultural University, Lanzhou 730070, China; hezh@st.gsau.edu.cn

**Keywords:** high altitude, hypoxia, physiological, genetic, adaptation

## Abstract

This review examines the remarkable adaptations of high-altitude indigenous animals to hypoxic environments. It discusses the physiological and biochemical strategies employed by these animals to enhance O_2_ uptake and delivery, as well as to increase the efficiency of O_2_ utilization. Key adaptations in the pulmonary and cardiovascular systems, including increased lung volume, efficiency of blood–O_2_ exchange, and remodeling of pulmonary vasculature, are highlighted. Additionally, the review explores adaptations in O_2_-consuming tissues, focusing on enhanced mitochondrial function and altered metabolic pathways. The role of genetic factors, particularly the hypoxia-inducible factor (HIF) pathway, is emphasized, showcasing the convergence of evolution across different species. The manuscript concludes by emphasizing the importance of further research integrating various omics approaches and studying multiple tissues and organs to fully understand the complex mechanisms of high-altitude adaptation.

## 1. Introduction

In nature, there is a great variety of organisms with different forms and their respective physiological functions. But any life phenomenon includes four basic characteristics, namely metabolism, excitability, reproduction, and adaptability. Researchers have long been fascinated by the adaptations that high-altitude indigenous animals exhibit in hypoxic environments. Studies have found that most people experience certain physiological, biochemical, anatomical, and clinical changes at altitudes over 3000 m, and some are even affected at 2000 m [1]. Generally, scientists define altitudes above 2500 m as “high altitude”, at which altitude most people display a fall in arterial O_2_ saturation (SaO_2_) [2]. Worldwide, the major high-altitude regions include the Qinghai-Tibet Plateau, the East African Plateau, and the Andean Plateau [3,4] (Figure 1). In the 1920s, studies on the hypoxic adaptation of high-altitude aborigines had already appeared, and for almost a century thereafter, the field of high-altitude adaptation was dominated by studies on humans [5,6,7,8]. In recent decades, however, studies of high-altitude adaptation in other species that live in a common home with these aborigines have also proliferated [9,10,11,12,13,14,15,16,17].

The partial pressure of O_2_ (PO_2_) decreases as the altitude increases; at an altitude of 3000 m, the PO_2_ is less than 70% of that at sea level [18,19]. The resulting hypobaric hypoxia poses a harsh physiological challenge to animals surviving and reproducing in high-altitude areas, and over the long term, a series of adaptive changes, or even damages, to their tissues and organs will occur. Therefore, high altitude is an ideal natural laboratory for studying human and animal adaptation to hypoxic environments.

Both humans and animals inhabiting high altitudes must evolve effective adaptive strategies to cope with limited O_2_ availability if they are to carry out normal survival and reproduction. These strategies are manifested in two main aspects: firstly, adaptive remodeling of the pulmonary and cardiovascular systems to enhance O_2_ uptake and delivery; and secondly, adaptive remodeling of O_2_-consuming tissues to increase the efficiency of O_2_ use and to reduce O_2_ consumption (Figure 2). These strategies are accomplished based on changes in the organism at the physiological and biochemical levels, which in turn are based on changes at the molecular genetic level.

This review aims to provide a comprehensive understanding of the remarkable physiological and genetic adaptations that enable high-altitude indigenous animals to thrive in hypoxic environments. By exploring the intricate interplay between changes in the pulmonary and cardiovascular systems, O_2_-consuming tissues, and genetic factors, we hope to shed light on the complex mechanisms underlying high-altitude adaptation. This review will not only focus on the key adaptations observed in hypoxia-sensitive tissues and organs but also delve into the molecular genetic mechanisms, particularly the Hb and HIF pathways, that contribute to these adaptations. Furthermore, we will discuss the broader implications of these adaptations, not only for understanding human high-altitude adaptation but also for exploring the fundamental principles of life in extreme environments.

## 2. High-Altitude Hypoxic Adaptation and Physiological Biochemical Characteristics

### 2.1. Pulmonary and Cardiovascular System Characteristics

Multicellular animals can respond to hypoxia, and the adaptive changes that occur in the pulmonary and cardiovascular systems are some of the most remarkable responses in mammals. The lungs are in the thoracic cavity and include both the left and right lungs, and their most important functions are ventilation and gas exchange. Pulmonary ventilation is the process of exhaling waste gases from the lungs out of the body and inhaling air into the lungs, completing the exchange of gases between the lungs and the environment. Pulmonary gas exchange is the process of O_2_ in the air being inhaled into the alveoli across the air-blood barrier into the capillaries and binding with hemoglobin (Hb), while CO_2_ from the capillaries enters the alveoli, completing the gas exchange between the alveolar air and the capillaries. The air-blood barrier, also known as the respiratory membrane, consists of a six-layered structure between alveolar gas and capillary blood [20].

Low PO_2_ in high-altitude environments leads directly to a decrease in arterial PO_2_ in animals, namely hypoxemia. The first physiological response to environmentally low PO_2_ is to increase pulmonary ventilation to minimize the fall in arterial PO_2_ [21]. This is generalizable to hypoxia-exposed low-altitude animals as well as high-altitude indigenous animals, the former of which experience an increase in pulmonary ventilation within the first few minutes of hypoxic exposure [22]. High-altitude indigenous animals such as Tibetan sheep and deer mice (*Peromyscus maniculatus*) exhibit increased resting respiratory rate and alveolar ventilation efficiency compared to their low-altitude counterparts [23,24]; one of the reasons for this is the adaptive changes in their pulmonary ventilation chemoreflexes [25]. Alterations in lung structure and function are also important for changes in pulmonary ventilation. Studies have shown that Quechua females living in the Andean Plateau have larger lung volumes than their lower altitude counterparts [26], and that larger lungs require correspondingly larger thoracic cavities, as has been found in high-altitude populations and in yaks (*Bos grunniens*) [27,28]. In addition, increased alveolar number and area have been found in many high-altitude indigenous animals, such as guinea pigs (*Cavia porcellus*) [29], dogs (*Canis lupus familiaris*) [30], Andean geese (*Chloephaga melanoptera*) [31] and yaks [32], etc.

An increase in the number and area of the alveoli implies an increase in the total surface area for blood–O_2_ exchange. However, for adequate blood–O_2_ exchange to occur, it also requires a correspondingly well-developed pulmonary vascular network effectively matched to the alveolar network to ensure that the O_2_ supplied by increased pulmonary ventilation can diffuse into the bloodstream through pulmonary gas exchange and be transported throughout the body by contraction of the left ventricle. As expected, it was found that the lungs of Andean geese living at 3000–5500 m altitude showed significant vascularization [31], which contributes to adequate blood–O_2_ exchange. Also, lung ventilation–blood perfusion matching during chronic hypoxia like in deer mice is more effective than in their lower altitude counterparts [33]. Increased alveolar area and pulmonary vascular abundance provide intact sites where more blood–O_2_ exchange can take place, and the thinning of the air–blood barrier comprising both allows for increased efficiency of blood–O_2_ exchange, as in the lungs of yaks [34].

In addition to increasing the abundance of the pulmonary vasculature, hypoxia in animals also promotes remodeling and some degree of muscularization of the pulmonary vasculature [35,36,37]. This is because hypoxia disrupts the integrity of the vascular endothelium and triggers the inward flow of growth factors, leading to smooth muscle cell proliferation and pulmonary artery thickening [38]. Nevertheless, there are contrary and at the same time rare examples, such as the thinner walls of the pulmonary arteries in mountain viscachas (*Lagidium peruanum*), which mean that the hypoxic adaptations of this species must be understood in combination with other tissues, organs, and systems [39]. Arterial wall thickening and smooth muscle cell proliferation result in increased vasodilation and vasoconstriction capacity, which is crucial for animals inhabiting plateaus because, in addition to increasing pulmonary ventilation in the face of low PO_2_, the second physiological response is to increase blood flow. Studies have shown that yaks have larger hearts than cattle [40], Tibetan pigs have thicker heart walls and richer intermuscular vasculature [41], Sherpas and Andean natives have increased plasma volume [42], and Han Chinese heart rate increases with altitude during plateau exposure [43], although their cardiac output, heart working capacity, and other indicators are lower than those of the Tibetan population [44]. These changes in the heart increase blood flow and blood pressure at the same time, which can lead to a common mountain sickness: pulmonary hypertension. The good vasodilatation and vasoconstriction capacity of the pulmonary artery can accommodate the large volume of blood pumped by the right ventricle and propel it throughout the lungs through stored elastic potential energy, thus avoiding or alleviating pulmonary hypertension.

Another strategy for coping with low PO_2_ is changes in Hb. It is generally accepted that an increase in Hb concentration offsets a decrease in arterial SaO_2_, so that Hb concentration increases with altitude. This trend has been shown, for example, in lowland sojourners at high altitude [45], Andean natives [46], pigs (*Sus scrofa domestica*) [47], dogs [48] and South American camelids such as llamas (*Lama glama*) and alpacas (*Vicugna pacos*) [49,50]. However, it has been shown that at high altitudes, increased Hb concentrations do not enable animals to reach the level of maximal O_2_ uptake (VO_2_max) that their counterparts would have at sea level [51]. And many mammals do not show a linear relationship between Hb concentration and altitude, e.g., Tibetan horses and Tibetans have similar Hb concentrations to their low-altitude counterparts [13,52]. It may seem counterintuitive, but Tibetans generally have superior aerobic capacity compared to lowland sojourners who have been acclimatized to high-altitude environments [53]. This suggests that an increase in arterial SaO_2_ by increasing Hb concentration is not universal, i.e., different species have different mechanisms to increase arterial SaO_2_. Recent studies have shown that Tibetans have a significant increase in plasma volume, which allows an increase in total Hb quantity without elevating Hb concentration [42]. The increased total Hb quantity increased O_2_ content in the arteries, whereas Hb concentrations similar to those of low-altitude Han Chinese avoided an increase in blood viscosity, thus reducing stress on the heart and damage to the blood microcirculation. These results demonstrate the importance of integrating multiple adaptive phenotypes at higher levels rather than emphasizing a single trait in isolation.

O_2_ content in the arteries is also affected by the quality of Hb (O_2_ affinity), in addition to the quantity of Hb. Many high-altitude indigenous animals exhibit higher Hb–O_2_ affinity [49,54], and one of the reasons for this increased affinity is due to amino acid substitutions caused by genetic variation in Hb subunit genes. For example, Tibetan mastiffs have acquired two specific amino acid substitutions on the Hb β-chain from the gene introgression of the Tibetan wolves, resulting in an increased Hb–O_2_ affinity, so that the Hb of Tibetan wolves and Tibetan mastiffs is distinct from that of their low-altitude counterparts [48,55]. All three nucleotide substitutions on the β-chain of plateau pika (*Ochotonidae*) Hb resulted in an increase in Hb–O_2_ affinity and showed a significant epistasis [56]. Genetic variants in the Hb subunit genes of the plateau deer mice also resulted in increased Hb–O_2_ affinity, meanwhile suppressing sensitivity to allosteric coenzyme factors such as Cl^−^ [57]. In addition, genetic variation in the Hb subunit genes has also resulted in increased Hb–O_2_ affinity in some birds [58,59].

The composition of Hb by different subunits is another factor affecting the affinity of Hb–O_2_. Studies have shown that the family of genes encoding Hb subunits is developmentally regulated, i.e., structurally and functionally distinct Hb isoforms are expressed at different developmental stages. In general, the order of affinity with O_2_ is fetal Hb isoform (HbF) > children Hb isoform (HbC) > adult Hb isoform (HbA) [60]. Our unpublished data show that adult Tibetan sheep have higher Hb concentrations than lambs; this may be due to the lower Hb–O_2_ affinity of HbA, which needs to be compensated by increased concentrations. It has also been shown that when adult sheep (*Ovis aries*) and goats (*Capra hircus*) are acutely exposed to hypoxia, they decrease HbA and increase HbC in response to low PO_2_ [61,62]. In addition, Tibetan antelope (*Panthelops hodgsonii*) exhibit an extreme case: complete loss of the adulthood-expressed hemoglobin subunit beta gene (*HBB*), resulting in the failure of the HbA to form. The HbC continues to be expressed in adulthood, resulting in the HbC becoming the only Hb isoform expressed by adult Tibetan antelope, which ensures the aerobic capacity of Tibetan antelope [10].

In conclusion, many studies have shown that animals coping with arterial low PO_2_ due to ambient low PO_2_ enhance uptake and delivery of O_2_ by increasing pulmonary ventilation, area and efficiency of blood–O_2_ exchange, blood flow, vasodilatation and vasoconstriction capacity, total Hb quantity, and Hb–O_2_ affinity. On the other hand, it also increases the efficiency of the use of scarce O_2_ by remodeling O_2_-consuming tissues.

### 2.2. Characteristics of O_2_-Consuming Tissues

The continuous delivery of sufficient O_2_ to each cell that makes the organism meet the metabolic demands of these cells is one of the necessary elements for the survival of animals. Animals inhabiting high-altitude hypoxic environments are faced with the paradox of O_2_ supply being less than demand, in which case the cells tend to remodel the O_2_-consuming tissues in order to increase the efficiency of O_2_ utilization and to reduce O_2_ consumption [63,64]. Mitochondria produce energy through the tricarboxylic acid cycle and oxidative phosphorylation (OXPHOS) and are the ultimate consumers of O_2_ and metabolic fuels. High-altitude hypoxic environments cause a persistent inhibition of OXPHOS and based on data from the muscles of some high-altitude indigenous animals, it has been proposed that enhanced aerobic capacity or intracellular redistribution of mitochondria could partially counteract the effects of low PO_2_ [65,66]. The number and cristae area of mitochondria in the muscle are proportional to aerobic capacity, and mitochondria undergo adaptive changes when aerobic capacity is inhibited [67].

It was found that the number and cristae area of mitochondria in several organs were greater in Tibetan sheep than in sheep breeds at lower altitudes [23]. The gastrocnemius muscle of high-altitude deer mice has a higher proportion of oxidized fibers and has evolved a greater respiratory capacity [68]. The increase in the number and cristae area of mitochondria may be the main reason for the greater respiratory capacity of the gastrocnemius muscle, and these increased mitochondria are mainly enriched in the subsarcolemmal [69]. Subsarcolemmal enrichment of mitochondria has also been found in the pectoral muscles of bar-headed geese (*Anser indicus*, which fly over the Himalayas during migration) [70]. Studies in humans have shown that preferential mitochondrial proliferation in the subsarcolemmal correlates with an increased aerobic capacity [71], and an increase in mitochondrial volume and density occurs when plains people stay at an altitude of 3454 m for 28 days [72]. The opposite phenomenon was observed in people who had been at extremely high altitudes (>5100 m) for a long period of time [73], and a significant reduction in skeletal muscle mitochondrial density was seen in returnees after summiting Mount Everest [74]. Enrichment in the subsarcolemmal brings mitochondria closer to O_2_ in the capillaries, which shortens the distance of O_2_ diffusion into the mitochondria, improves the efficiency of O_2_ transport, and helps to maintain the supply of O_2_ to the mitochondria in hypoxic environments. And the increase in mitochondrial number and cristae area provides more attachment sites for OXPHOS-related substrates and enzymes and leads to an increase in the organism’s OXPHOS capacity, which implies a higher O_2_ utilization capacity. The reason for the decrease in mitochondrial density after high-altitude exposure in some populations may be that the exposure was too severe, leading to the manifestation of this maladaptive response.

Another strategy to increase the efficiency of O_2_ utilization is to partially change the substrate of energy metabolism, i.e., to decrease fatty acid oxidation (FAO) and increase carbohydrate oxidation. Because carbohydrate oxidation produces more adenosine triphosphate per mole of O_2_ consumed than FAO [75], this change in metabolic substrate preference has been found in many high-altitude indigenous animals. For example, Tibetans have higher serum levels of non-esterified fatty acids, which suggests that Tibetans may down-regulate FAO [76], and Sherpas have reduced expression of Peroxisome proliferator-activated receptor alpha (*PPARα*) and its target gene Carnitine palmitoyltransferase 1B (*CPT1B*) in skeletal muscle, which can lead to a decreased FAO capacity of mitochondria [77]. In addition, when high-altitude deer mice exercise at 75% of VO_2_max, the proportion of carbohydrate oxidation increases, whereas this is not the case in low-altitude deer mice [78]. Similar differences in metabolic substrate selection preferences during exercise have been observed between Andean leaf-eared rats (*Phyllotis xanthopygus*) and their low-altitude counterparts [79]. This may facilitate the maintenance of muscle performance at low PO_2_.

In addition to increasing the efficiency of O_2_ utilization by increasing the number and cristae area of mitochondria, changing the morphology and distribution of mitochondria, and altering the selection preference of energy metabolism substrates, animals inhabiting high altitudes also reduce O_2_ consumption by a moderate shift in the mode of energy metabolism from aerobic to anaerobic fermentation. Increased glucose (Glu) metabolism, particularly glycolysis, is a hallmark of adaptation to high-altitude hypoxia [80]. It was found that not only FAO was down-regulated but pyruvate oxidation was also inhibited in hypoxic rat (*Rattus norvegicus*) hearts, suggesting increased glycolysis [81]. In addition, positively selected Egl-9 family hypoxia-inducible factor 1 (*EGLN1*) gene haplotypes were associated with elevated serum lactate levels in Tibetans [76], higher lactate dehydrogenase activity in the muscles of Sherpas than in low-altitude individuals, which resulted in enhanced lactate metabolism [77]. Furthermore, Glu uptake is higher in the hearts of Sherpas than in low-altitude populations [81]. These findings suggest that reducing O_2_ consumption by increasing glycolysis in high-altitude animals is also one of the strategies for adapting to high-altitude hypoxic environments.

### 2.3. Oxidative Stress

The balance between pro-oxidant and anti-oxidant activities in the organism is essential for normal life activities, and if this balance is disrupted and pro-oxidants dominate, oxidative stress can result. Exposure to hypoxia reduces the O_2_ supply to the cell and decreases the activity of cytochrome c oxidase, which transfers electrons to O_2_ in the mitochondria, thus affecting the redox balance and leading to the production of reactive O_2_ species (ROS), which accelerate accumulation with increasing altitude [82].

Mitochondria are the main site of ROS production, and in vitro studies have shown that approximately 0.1–2% of the O_2_ consumed by mitochondria ends up as ROS rather than combining with electrons delivered by cytochrome c oxidase to generate water [83]. Excessive ROS accumulation in cells and tissues leads to a variety of oxidative damages, but the beneficial aspect is that increased generation of ROS (especially produced by complex III) may play an important signaling role in hypoxic environments. For example, stabilizing the hypoxia-inducible factor subunit 1 alpha (HIF-1α) protein promotes the expression of a variety of downstream hypoxia-responsive genes [84], which in turn negatively feedback reduces ROS production. This negative feedback regulation may have resulted in reduced ROS production in Tibetans and Sherpas relative to lowland sojourners [77,85], which in turn attenuates oxidative stress in organism tissues and organs. In addition, reduced ROS production was also observed in mitochondria isolated from the hindlimb muscles of deer mice that were well adapted to the hypoxic environment [69].

Meanwhile, ROS release from the mitochondria of the diaphragm was increased after hypoxic exposure in low-altitude deer mice. This could be due to the hypoxic environment having a greater effect on ROS production in hypoxic exposed low-altitude deer mice [86]. Thus, in chronic hypoxic environments, high-altitude indigenous animals may modulate mitochondrial ROS production to alter cell signaling and attenuate oxidative stress in cells and tissues.

### 2.4. Other Systems Related to Adaptation to High-Altitude Hypoxic Environments

In addition to the pulmonary, cardiovascular, and O_2_-consuming tissues described above, other systems such as the endocrine system, the central nervous system, and the immune system, as well as their closely related endocannabinoid systems (ECs), also play crucial roles in mediating the complex adaptive responses to hypoxia. The ECs, a complex network of lipid signaling molecules, receptors, and metabolic enzymes, have emerged as a potential mediator of the adaptive response to hypoxia. The ECs can bind to cannabinoid receptors (CB1 and CB2), which are widely distributed throughout the body and regulate various physiological processes [87]. Recent studies have increasingly shown that the ECs are also involved in the process of hypoxia adaptation. For example, the levels of N-acylethanolamides (NAEs) in the blood of residents living at high altitudes are significantly higher than those in residents living at low altitudes and are associated with increased hemoglobin concentration. This suggests that NAEs, particularly palmitoylethanolamide and oleoylethanolamide, that modulate the ECs, may be involved in the physiological regulation after long-term exposure to high altitudes, such as the increase in erythrocytosis and the enhancement of O_2_ transport capacity [88]. In addition, research shows that endurance and resistance exercise can regulate the levels of ECs and receptors in the ECs, as well as the downstream signaling pathways, indicating that the ECs may be involved in the adaptation to exercise in hypoxic environments. However, the regulation of ECs by exercise may differ under normoxic and hypoxic conditions, and its physiological effects need further investigation [89].

Besides exercise, the cannabinoid agonist can reduce the levels of early inflammatory factors after hypoxia-ischemia, which may help alleviate neuroinflammation and improve brain damage [90]. In addition, the ECs involved in regulating cerebral blood flow include the influence of CB1 and CB2 receptors and transient receptor potential vanilloid type 1 channels on the activity of smooth muscle cells, endothelial cells, and neurons, as well as the regulation of inflammatory reactions. This suggests that the ECs may improve cerebral perfusion under hypoxic conditions by regulating cerebral blood flow [91]. Furthermore, research finds that prenatal exposure to the cannabinoid agonist can lead to increased ventilation, altered responses to hypoxia, and longer apnea in newborn mice, indicating that the ECs may be involved in the adaptation of newborns to hypoxia [92]. Finally, research shows that CB1 receptor antagonists can improve glucose transport activity in the skeletal muscle of insulin-sensitive and insulin-resistant rats, while CB1 receptor agonists have the opposite effect, suggesting that the ECs may be involved in the regulation of energy metabolism under hypoxic conditions [93].

In summary, the ECs play a multifaceted role in hypoxia adaptation, including the regulation of erythropoiesis, exercise adaptation, neuroinflammation, cerebral blood flow, and energy metabolism. However, the specific mechanisms and target sites of the ECs in hypoxia adaptation still need to be further studied to provide new ideas and targets for the prevention and treatment of hypoxia-related diseases.

## 3. Genetics Study of Hypoxia Adaptation at High Altitude

Humans and animals have long survived and thrived in high-altitude environments around the world, and these high-altitude indigenous species have evolved good adaptations to hypoxia. Different species and even different populations of the same species (e.g., Tibetans and Andean natives) show differences in some physiological and biochemical phenotypes (e.g., Hb). However, when multiple adaptive phenotypes were comprehensively considered at a higher level, they showed consistency. At the genetic level, these species also show some degree of convergent evolution, with specific genes and molecular pathways involved in adaptation to hypoxia in multiple species (Table 1). For example, the endothelial PAS domain protein 1 (*EPAS1*) gene encoding HIF-2α and the HIF pathway it participates in are frequent targets of selection in hypoxic environments.

Genome-wide comparative studies of Tibetans and Han Chinese have revealed that four genes in the HIF pathway, *EGLN1*, *EPAS1*, *PPARα*, and Heme oxygenase 2 (*HMOX2*), are positively selected [94,95,96], and that mutations at certain loci of these genes affect physiological and biochemical phenotypes associated with high-altitude hypoxic adaptation. For example, *EGLN1* encodes prolyl hydroxylase domain protein 2 (PHD2), which acts to degrade HIF-α in normoxic environments, whereas hypoxia inhibits the activity of PHD2, leading to the stable expression of HIFs and initiating a series of hypoxia physiological responses such as erythropoiesis [18]. Two missense mutations in the *EGLN1* gene in Tibetans elevate the activity of PHD2, which also degrades HIF-α in a hypoxic environment, resulting in Tibetans showing a blunted response to erythropoiesis and protecting them from erythrocytosis [97]. Sequence variants in *EPAS1*, *PPARα*, and *HMOX2* genes are significantly associated with blood traits such as low Hb concentration [94,98]. Genome-wide studies of Andean natives identified positively selected genes such as *EGLN1*, endothelin receptor type A (*EDNRA*), protein kinase AMP-activated catalytic subunit alpha 1 (*PRKAA1*), and nitric oxide synthase 2A (*NOS2A*) [96,99], with the *EDNRA* and *PRKAA1* genes associated with greater birth weight and uterine artery diameters [100]. A genomic study of Amhara and Oromo populations living in the East African Plateau found that basic helix-loop-helix family member e41 (*BHLHE41*), capicua transcriptional repressor (*CIC*), lipase E (*LIPE*), platelet-activating factor acetylhydrolase 1b catalytic subunit 3 (*PAFAH1B3*), and endothelin receptor type B (*EDNRB*) genes were positively selected in both populations, with *BHLHE41* being involved in the initiation of hypoxia response through the HIF pathway, while the latter four were associated with enhanced hypoxic tolerance [101,102,103].

In addition, the *EPAS1* and *HBB* genes were found to be positively selected in Tibetan mastiffs, and variants appearing on these genes were associated with reduced blood flow resistance and increased Hb–O_2_ affinity, respectively [48], which are thought to be the result of gene introgression from Tibetan wolves [55]. Another study on the X chromosome of Tibetan mastiffs found that the angiomotin (*AMOT*) gene was also targeted for selection and was associated with blood pressure regulation [104]. The *EPAS1* gene was also positively selected in Tibetan cashmere goats, in addition to several genes identified as being associated with hypoxic adaptation [105]. Studies in high-altitude deer mice have found allelic variation in *EPAS1* to be associated with cardiovascular function and transcriptional responses to hypoxia [106]. Two missense mutations in the *EPAS1* gene of the Tibetan horse were closely related to the promotion of blood circulation and O_2_ transport [13]. In addition, the mitochondrial genome of the Tibetan horse is also subjected to selection by hypoxic environments, with a high rate of non-synonymous mutations in the NADH dehydrogenase subunit 6 (*NADH6*) gene, which suggests that changes in energy metabolism are an important aspect of the adaptation of the Tibetan horse to the high-altitude hypoxic environments [107]. As an important gene upstream of the HIF pathway, a specific allele of *EGLN1* of yak genome introgression into Tibetan cattle may reduce Hb concentration and hematocrit of the latter [108]. Other positively selected genes identified in yaks are ADAM metallopeptidase domain 17 (*ADAM17*) and arginase 2 (*ARG2*), which are involved in hypoxic stress response [109]. Studies on Tibetan chickens identified genes involved in calcium signaling pathways, such as ryanodine receptor 2 (*RYR2*), which may be associated with hypoxia tolerance [110].

Different species inhabiting high-altitude environments show convergent evolution, and even the closely related species may gradually close the genetic distance due to gene exchange caused by hybridization (e.g., yak vs. cattle, Tibetan mastiff vs. Tibetan wolf). However, there are still unique selective features within the same species, which correspond to unique adaptive mechanisms. For example, many hypoxia-associated genes are positively selected for in several pig breeds on the Qinghai-Tibet Plateau, and variation in some of these genes (e.g., regulator of the cell cycle, *RGCC*) is shared across breeds, whereas variation in others (e.g., glutamate ionotropic receptor NMDA type subunit 2B, *GRIN2B*) is unique to a particular breed [111]. Different targets of selection have also been identified in different sheep breeds on the Qinghai-Tibet Plateau, e.g., the suppressor of cytokine signaling 2 (*SOCS2*) gene was identified in Tibetan sheep and is associated with energy metabolism [112]. The fibroblast growth factor 7 (*FGF7*) gene was identified in Nepalese sheep, and one of its upstream variants was associated with inhibition of hypoxia-induced lung injury [113]. This suggests that in addition to convergent evolution at the molecular genetic level, animals also respond to the hypoxic challenge through unique genetic variations.

## 4. Conclusions

High-altitude aborigines and indigenous animals are better adapted to hypoxic environments than sojourners from low altitudes. Undoubtedly, the mechanism of this adaptation is extremely complex, but it is mainly manifested in the following aspects: (1) adaptive remodeling of the pulmonary and cardiovascular systems to enhance the capacity of O_2_ uptake and delivery, such as increasing pulmonary ventilation and blood flow, and total Hb and Hb–O_2_ affinity; (2) adaptive remodeling of O_2_-consuming tissues to increase the efficiency of O_2_ use, such as increasing mitochondrial number and carbohydrate oxidation; (3) positive selection of key genes and pathways, especially those belonging to the HIF pathway. In the future, as technology advances, various omics, from genomics to phenomics, as well as multiple tissues and organs and even cross-species studies, will have to be integrated to fully understand what is happening in the extraordinary natural laboratory that is high altitude.

## Figures and Tables

**Figure 1 animals-14-03031-f001:**
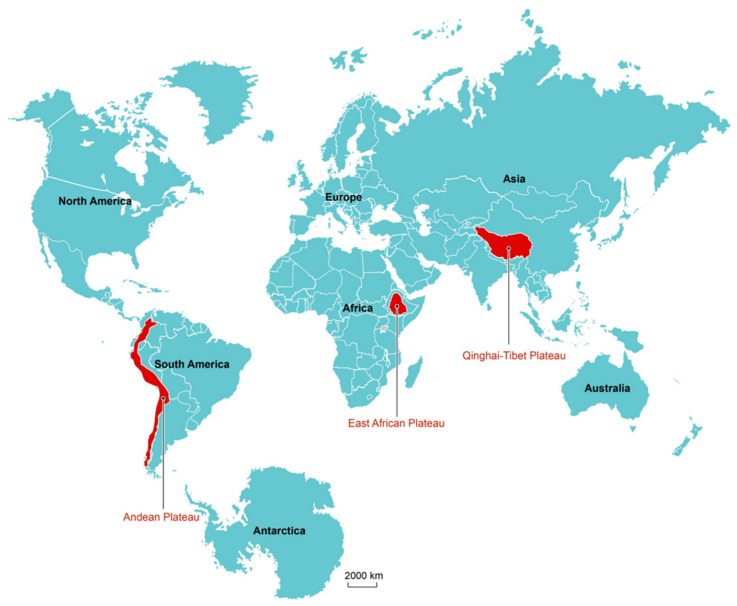
The geography of high-altitude indigenous animals’ adaptation to hypoxic environments. The geographic locations where indigenous animals have adapted to life at high altitudes are in red and include (from right to left) the Qinghai-Tibet Plateau, the East African Plateau, and the Andean Plateau.

**Figure 2 animals-14-03031-f002:**
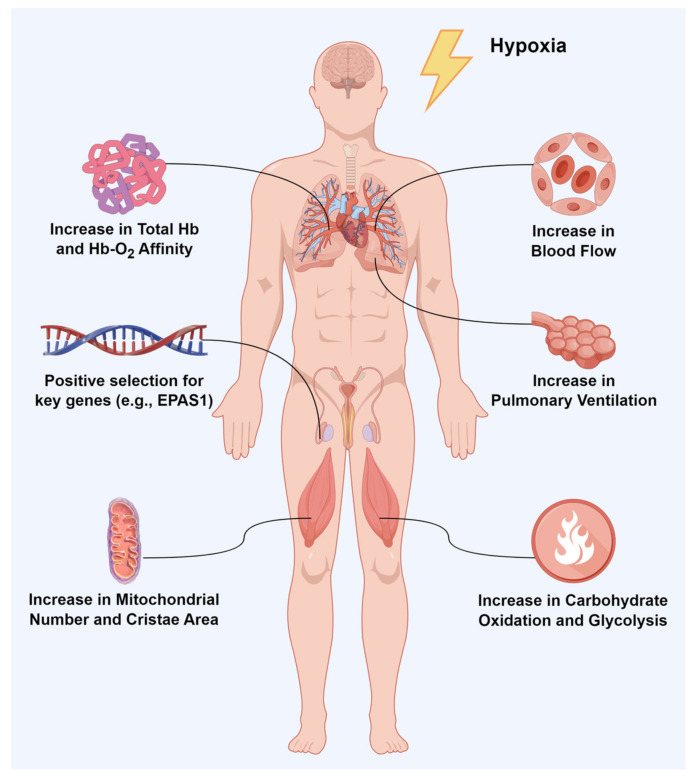
In response to hypoxia, the organism undergoes adaptive changes at both physiological and biochemical levels and molecular genetic levels.

**Table 1 animals-14-03031-t001:** Positively selected genes and their functions in indigenous species on the Qinghai-Tibet Plateau.

Research Object	Positively Selected Genes	Main Functions of These Genes in High-Altitude Hypoxia Adaptation
Tibetans [94,95,96,97,98]	*EGLN1*, *EPAS1*, *PPARα*, *HMOX2*	Degraded HIF-α (*EGLN1*); Associated with blood traits (*EPAS1*, *PPARα*, *HMOX2*)
Andean natives [96,99,100]	*EGLN1*, *EDNRA*, *PRKAA1*, *NOS2A*	Associated with greater birth, weight, and uterine artery diameters (*EDNRA*, *PRKAA1*)
Amharas and Oromos [101,102,103]	*BHLHE41*, *CIC*, *LIPE*, *PAFAH1B3*, *EDNRB*	Involved in the initiation of hypoxia response through the HIF pathway (*BHLHE41*); Associated with enhanced hypoxic tolerance (*CIC*, *LIPE*, *PAFAH1B3*, *EDNRB*)
Tibetan mastiffs [48,55,104]	*EPAS1*, *HBB*, *AMOT*	Reduced blood flow resistance (*EPAS1*); Increased Hb–O_2_ affinity (*HBB*); Blood pressure regulation (*AMOT*)
Tibetan cashmere goats [105]	*EPAS1*	Hypoxia adaptation
Deer mice [106]	*EPAS1*	Associated with cardiovascular function and transcriptional responses to hypoxia
Tibetan horse [13,107]	*EPAS1*, *NADH6*	Promoted blood circulation and O_2_ transport (*EPAS1*); Associated with energy metabolism (*NADH6*)
Yak [108,109]	*EGLN1*, *ADAM17*, *ARG2*	Reduced Hb concentration and hematocrit (*EGLN1*); Involved in hypoxic stress response (*ADAM17*, *ARG2*)
Tibetan chickens [110]	*RYR2*	Associated with hypoxia tolerance
Tibetan pig [111]	*RGCC*, *GRIN2B*	Involved in hypoxia-induced anti-angiogenesis (*RGCC*); Neural response (*GRIN2B*)
Tibetan sheep [112,113]	*SOCS2*, *FGF7*	Associated with energy metabolism (*SOCS2*); Inhibition of hypoxia-induced lung injury (*FGF7*)

## Data Availability

Not applicable.

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
