# Peer review of "Physiological and Genetic Basis of High-Altitude Indigenous Animals’ Adaptation to Hypoxic Environments"

_animals, 2024, doi:10.3390/ani14203031_

Round 1

Reviewer 1 Report

Comments and Suggestions for Authors

This review paper titled Physiological and Genetic Basis of High-Altitude (HA) Indigenous Animals Adaptation to Hypoxic Environments addresses an interesting line of analysis to understand the phenomena of adaptation or lack of adaptation to living at high altitudes.

Although the objective of the study has been described in the abstract indicating that this paper reviews the adaptive changes in hypoxia-sensitive tissues and organs, as well as at the molecular genetic level, such as pulmonary, cardiovascular, oxygen-consuming tissues, and the Hb and HIF pathway, that occur in animals in response to the challenge of hypobaric hypoxia, the introduction section lack to describe the objectives of the review paper.

However, the objective is not explicitly described in the text at the end of the introduction section. Authors should include a paragraph at the end of the introduction section indicating the objectives of the review paper.

Authors should include a section about the strategies used by humans and animals for adaptation to living at HA how they are similar (animals vs humans) and how are different.

Adaptation is a process in which is involves the endocrine system, the Central Nervous System (CNS), and the Immunology systems. These in turn may be linked to the endocannabinoid system activity. The endocannabinoid system has been considered an adaptogen and several papers have been focused on EC and Hb production at high altitudes.

It is suggested that the authors include a text on the interrelationship between these three important systems for adaptation and not only what they have expressed in the text.

Erythrocytosis and excessive erythrocytosis (EE) have been related to increased androgen activity ad and increased EPO activity. Serum T increases Hb and it is responsible for sexual dimorphism with higher Hb levels in men than in women. E has been related to chronic mountain sickness, and the authors must include a subtitle related to maladaptation to HA. In lines 42-45, the authors say that in the 1920s, studies on the hypoxic adaptation of high-altitude aborigines had already appeared, and for almost a century thereafter, the field of high-altitude adaptation was dominated by studies on humans. Interestingly almost 100 years ago, scientist Carlos Monge Medrano described for the first time on 2 June 1924, the first case of a subject with chronic mountain sickness. The text of this case was presented to the Peruvian National Academy of Medicine, and then, in 1925 was published in Peru, and then in 1929 in Paris, France was named “Monge´s disease”. This finding should be highlighted in this review article.

Another mechanism of adaptation that should be highlighted and that in the second decade of the last century was described by Sir Joseph Barcroft is that fetuses, including humans, develop in the womb as if they were living on Mount Everest (8800 m). The human fetus develops in a profoundly hypoxic environment. Then, the foundations of our physiology are built in the uterus under the most hypoxic conditions that we are ever likely to experience. This magnitude of exposure to hypoxia in utero is rarely experienced in adult life, with few exceptions. The fetal life in hypoxic conditions is necessary since neonates require a high amount of iron stores.  The only way to store a high amount of iron during pregnancy is in hypoxic conditions.

Indeed, the lowest recorded levels of arterial oxygen in adult humans are like those of a fetus and were recorded just below the highest attainable elevation on the Earth's surface: the summit of Mount Everest. 

It is interesting to address animal species that for longer periods than humans may have settled in altitudinal areas. There are, however, some omissions. For example, South American camelids such as llamas, alpacas, and vicuñas that live for a long time at altitude have not been addressed. In the scientific literature, there is an important production of articles dedicated to these camelids. One species adapted to highlands in South America belongs to the sub-order Hystricomorphs which include the guinea pigschinchillas, and mountain viscachas. These have not been included in the review paper, except for some information on guinea pigs.

According to the authors, both humans and animals inhabiting high altitudes must evolve effective adaptive strategies to cope with limited O2 availability if they are to carry out normal survival and reproduction. These strategies are manifested in two main aspects:

1)    Adaptive re-modeling of the pulmonary and cardiovascular systems to enhance O2 uptake and delivery; and

2)    Adaptive remodeling of O2-consuming tissues to increase the efficiency of O2 use and to reduce O2 consumption.

These strategies are accomplished based on changes in the organism at the physiological and biochemical levels, which in turn are based on changes at the molecular genetic level. However, the role of other systems (immune, endocrine, and nervous) in achieving these changes is not emphasized and they are lacking in the review paper.

There are several strategies that the body uses to have an adequate transport of oxygen to the cells, and these should be described in detail:

1.            Raise the Hb level with the right shift of the oxy-hemoglobin curve (low affinity).

2.            Failure to moderately increase or increase Hb level with a left shift of the oxy-hemoglobin curve (high affinity)

3.            Increase Hb and a greater increase in plasma volume as in Sherpas.

High-altitude aborigines and indigenous animals are better adapted to hypoxic environments than sojourners from low altitudes. However, there is not a universal pattern of adaptation. Han and Tibetans living in the same place in Tibet have different Hb levels. Adaptation in the Andean region is better in the southern region of the Andes, and there is less adaptation in the central Andes.

Undoubtedly, the mechanism of this adaptation is extremely complex, but it is mainly manifested in the following aspects: 1) Adaptive remodeling of the pulmonary and cardiovascular systems to enhance the capacity of O2 uptake and delivery, such as increasing pulmonary ventilation and blood flow, total Hb and Hb-O2 affinity; 2) Adaptive remodeling of O2-consuming tissues to increase the efficiency of O2 use, such as increasing mitochondrial number and carbohydrate oxidation; 3) Positive selection of key genes and pathways, especially those belonging to the HIF pathway.

It is important to put a figure with a shift of the oxy-hemoglobin curve indicating at what altitude, there is a dramatic change in oxygen saturation. From 0 to 2500 m, almost no change in the oxy-hemoglobin curve is observed. At this point, it should be discussed why the WHO adjusted Hb concentration to define anemia since 500 m (-0.4 g/dl of Hb). This strategy to adjust Hb for altitude to define anemia at the highlands increased the prevalence of anemia at high altitudes. Authors should discuss oxygen transport at different altitudes and if only Hb reflects oxygen transport.  

Reviewer 2 Report

Comments and Suggestions for Authors

The authors present a detailed review about physiological and genetic basis of high-altitude indigenous animals’ adaptation to hypoxic environments. Generally speaking, the overall writing of this manuscript is good in a well-structured manner, and numerous details were compared to support the main conclusions. This topic is interesting and attractive to worldwide readers.

However, extra editing and minor revisions are required before acceptance for publication.

For example, Line 3 in the paper title: “Animals” should be “Animals’ ”; Line 10: change the comma to a period.

Provide the full name for any abbreviated term at its first appearance, such as Hb and HIF in lines 14-15, and EPAS1 in line 283.

Line 50: “blue” should be “red”?

Rewrite the Simple Summary, since it is almost the same as the Abstract.

The authors are recommended to draw a figure or provide a table to summarize molecular mechanisms for the high-altitude adaptations in various animals including human being.

By the way, recent publications in the past five years are recommended to cite.

Comments on the Quality of English Language

 Minor editing of English language is required.

Round 2

Reviewer 1 Report

Comments and Suggestions for Authors

I agree with revised version